# Effect of Gelling Agent Type on the Physical Properties of Nanoemulsion-Based Gels

Natalia Riquelme [1] , Constanza Savignones [1], Ayelén López [1], Rommy N. Zúñiga [2] and Carla Arancibia [1,*]

[1] Department of Food Science and Technology, Technological Faculty, Universidad de Santiago de Chile, Obispo Umaña 050, Estación Central 9170201, Chile; natalia.riquelme.h@usach.cl (N.R.)
[2] Department of Biotechnology, Universidad Tecnológica Metropolitana, Las Palmeras, 3360, Ñuñoa 7800003, Chile; rommy.zuniga@utem.cl
* Correspondence: carla.arancibia@usach.cl; Tel.: +56-(2)-27184518

**Abstract:** Senior populations may experience nutritional deficiencies due to physiological changes that occur during aging, such as swallowing disorders, where easy-to-swallow foods are required to increase comfort during food consumption. In this context, the design of nanoemulsion-based gels (NBGs) can be an alternative for satisfying the textural requirements of seniors. This article aimed to develop NBGs with different gelling agents, evaluating their physical properties. NBGs were prepared with a base nanoemulsion (d = 188 nm) and carrageenan (CA) or agar (AG) at two concentrations (0.5–1.5% *w/w*). The color, rheology, texture, water-holding capacity (WHC) and FT-IR spectra were determined. The results showed that the CA-based gels were more yellow than the AG ones, with the highest hydrocolloid concentration. All gels showed a non-Newtonian flow behavior, where the gels' consistency and shear-thinning behavior increased with the hydrocolloid concentration. Furthermore, elastic behavior predominated over viscous behavior in all the gels, being more pronounced in those with AG. Similarly, all the gels presented low values of textural parameters, indicating an adequate texture for seniors. The FT-IR spectra revealed non-covalent interactions between nanoemulsions and hydrocolloids, independent of their type and concentration. Finally, the CA-based gels presented a higher WHC than the AG ones. Therefore, NBG physical properties can be modulated according to gelling agent type in order to design foods adapted for seniors.

**Keywords:** nanoemulsion-based gels; agar; carrageenan; physical properties

## 1. Introduction

The increase in the size of the senior population (>60 years old) has become one of the most significant challenges for the food industry, which is required to design foods focused on this population group that respond to their sensory, biological, and nutritional requirements due to the physiological changes caused by aging [1]. For example, these individuals often have bad teeth or dental prostheses that decrease their chewing performance [2]. They also present changes in salivary discharge and food swallowing disorders, such as dysphagia, which can lead to dehydration, malnutrition, and aspiration pneumonia, reducing their quality of life [3,4]. For this reason, food textures for senior adults should be easy-to-swallow and moist, and the food should be easily disintegrated and mixed in the mouth, avoiding mastication with the teeth [5]. A promising alternative to texture-modified food focused on this population is developing gels, where a biopolymer-based network is built to retain water or colloidal dispersions [6]. Consequently, these gels are suitable for seniors, since they are ingested through compression between the tongue and the palate until their complete disintegration without any chewing processing, facilitating their swallowing [7]. Therefore, these matrices can be an excellent sensory alternative for older people, because these gels could be designed to provide pleasant consumption experiences [8].

Alternatively, the food industry has innovated itself through developing ingredients and functional foods using novel technologies, such as nanoemulsions [9]. Nanoemulsions are dispersions of two immiscible liquids, where the dispersed phase has a droplet size between 20 and 200 nm [10]. These systems allow for the carriage of bioactive lipid compounds and their incorporation into aqueous products [9]. In addition, they have several advantages related to conventional emulsions due to their smaller droplet size and increased interfacial area. For example, nanoemulsions can protect lipid compounds from interaction with other food components or unfavorable conditions [11], improve lipid compounds' bioaccessibility during digestion [12,13], and extend their kinetic stability [14], among others. For this reason, new nanoemulsified structures, such as nanoemulsion-based gels, can be used to develop products adapted to the oral requirements of older people, where different hydrocolloids and their mixtures can be applied to obtain easy-to-swallow foods for people with chewing or swallowing dysfunctions.

Nanoemulsion-based gels' structure can usually be formed in two steps, through preparing the nanoemulsions and then turning the nanoemulsions into gels [15]. This last stage can be performed through nanoemulsion lipid droplets, aggregation, forming a network structure, or through their dispersion into hydrocolloid dispersions [16]. In both cases, the food matrix corresponds to a combination of these different structures (the emulsion and gel), which have good physical stability and mechanical properties [17]. However, nanoemulsion-based gel properties depend on the interactions between oil droplets and hydrocolloids [15,18], which can make it difficult to elaborate on stable gel-based foods, since the interactions between ingredients affect their physical properties, especially the texture characteristics [19]. For this reason, selecting the hydrocolloid type is crucial for obtaining stable nanoemulsion-based gels with controlled rheological and textural properties focused on the needs of seniors, such as soft, moist, and easy-to-chew and -swallow characteristics [20]. In this sense, food-grade biopolymers, such as proteins and polysaccharides, are considered for nanoemulsion-based gels' preparation [21], where polysaccharides form a more stable structure in food products while proteins are prone to denaturation, limiting their applications.

Carrageenan and agar correspond to two biopolymers obtained from red marine algae, which are widely used in the food industry due to their numerous applications as thickening, gelling, and emulsifying agents [22]. In addition, they are considered a food additive and generally recognized as safe by the Food and Drug Administration and European Food and Safety Agency [23]. In the case of carrageenan, its structure is composed of a linear polysaccharide with alternating units of D-galactose and 3,6-anhydrous-galactose linked by α-1,3 and β-1,4 glycosidic bonds, which form a high-molecular-weight biopolymer [24]. In addition, among the different forms of carrageenan, κ-carrageenan is the most important one and can form gels in aqueous dispersions [25]. In the case of agar, it is a natural polysaccharide also extracted from various species of red algae. This polysaccharide consists of two fractions (agarose and agaropectin) in variable proportions, where agarose is responsible for the gelling properties [26], since it can form aggregates from compact and ordered helical structures [27], allowing for the formation of a firm or weak gel. The chemical structure of agar is composed of D-galactopyranosyl residues and 3,6-anhydrous-L-galactopyranosyl monomers linked together and alternating α-1,3 and β-1,4 bonds [23].

A few studies have been conducted on nanoemulsion-based gels, mainly related to their use as fat replacers [28–30] and their application as encapsulation systems of bioactive compounds [31–33]. However, the development of food products based on nanoemulsion-filled gels has not yet been reported. These food matrices may be ideal for the elaboration of easy-to-swallow desserts since these products are one of the most appreciated by seniors for their sensory properties and palatability [34]. Therefore, it is necessary to study how nanoemulsion-based gels can lead to a wide range of rheological and mechanical properties, which depend on the nature of the components (lipid phase, emulsifier, and hydrocolloid type and concentration) and their interactions [19]. In this context, this work aimed to evaluate the effects of hydrocolloid type and concentration on the physical properties

of nanoemulsion-based gels in order to obtain easy-to-swallow gels that can be used as potential foods for seniors with swallowing problems. In this sense, it is possible to hypothesize that using carrageenan and agar as gelling agents would allow one to obtain nanoemulsion-based gels with targeted physical properties, which could be an alternative for the development of food products such as puddings or custard desserts.

## 2. Materials and Methods

### 2.1. Materials

Nanoemulsions were prepared with canola oil (Belmont, Watt's S.A., San Bernardo, Chile) as a lipid phase; soy lecithin (Metarin P-Cargill, Blumos S.A., Santiago, Chile) and pea protein isolate (Nutralys® F85M, Roquette, Lestron, France) as emulsifiers; and purified water obtained from a reverse-osmosis system (Vigaflow S.A., Colina, Chile) as the aqueous phase. In addition, carrageenan (κ-carrageenan, Sabores.cl, Santiago, Chile) or agar (Tractor Bean, Santiago, Chile) was added into the nanoemulsions as gelling agents.

### 2.2. Preparation of Nanoemulsion-Based Gels

The base nanoemulsion preparation consisted of the following steps: (i) First, the aqueous phase was prepared by dispersing the pea protein (1% $w/w$) in purified water using a magnetic stirrer (Arex, Velp Scientifica, Usmate Velate, Italy) at 350 rpm for 40 min. Then, soy lecithin (3% $w/w$) was added to the aqueous phase and stirred for 40 min at 800 rpm until its complete dispersion. The aqueous phase was stored at $4 \pm 1$ °C in a glass beaker for 24 h. (ii) Subsequently, a coarse emulsion was prepared by dispersing the lipid phase (5% $w/w$ of canola oil) in the aqueous phase using a high-speed homogenizer (IKA T25, Ultra Turrax, Germany) at 10,000 rpm for 10 min. (iii) Finally, the pre-emulsion was subjected to a high-energy homogenization process using an ultrasound device (VCX500, Sonics, Orlando, FL, USA) to reduce the particle size on the nanometric scale. The process conditions were as follows: 20 kHz, 90% amplitude, and 19.5 min with 15 and 10 s work and rest intervals, respectively. The base nanoemulsion presented narrow monomodal droplet size distributions, and its mean droplet size was relatively small (Figure 1), with values equal to $188 \pm 1$ nm and a polydispersity index of 0.14, which were determined using Zetasizer (NanoS90, Malvern Instruments, Malvern, UK).

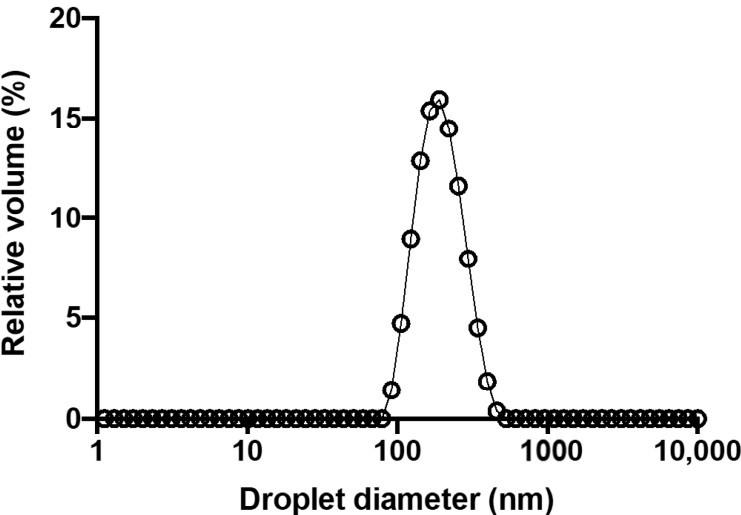

**Figure 1.** Particle size distribution of base nanoemulsion prepared with pea protein and soy lecithin as emulsifiers (1% and 3% $w/w$, respectively).

Finally, the nanoemulsion-based gels (NBG) were prepared by mixing the base nanoemulsion with different hydrocolloids (as gelling agent) at two concentrations: κ-carrageenan (0.5 and 1.5% $w/w$) and agar (1.0 and 1.5% $w/w$). These concentrations

were chosen according to preliminary work (data not show) to obtain NBGs with similar apparent viscosity at a shear rate of $10\,\mathrm{s}^{-1}$. This consideration was defined to obtain negligible distortion in the physical characterization results based on the effects of differences in the initial viscosity of the gels. The dispersion was carried out with a propeller stirrer (F20100151, Velp Scientifica, Usmate Velate, Italy) at 50 rpm in a thermoregulated bath (Heating bath B-100, Buchi, Flawil, Switzerland) at $80 \pm 1\,^{\circ}\mathrm{C}$ for 40 min. After that, the dispersions obtained were cooled at room temperature ($25 \pm 1\,^{\circ}\mathrm{C}$) and stored at $4 \pm 1\,^{\circ}\mathrm{C}$ for 24 h until further characterization. Each formulation was prepared in duplicate.

### 2.3. Optical Properties of Nanoemulsion-Based Gels

The optical properties of the nanoemulsion-based gels were determined using a colorimeter (ChromaMeter CR-410, Konica Minolta, Osaka, Japan), which was calibrated using a standard calibrated plate (L*: 93.46, a*: 0.42 and b*: 4.08). A 20 g piece of each sample was deposited in a Petri dish (60 mm internal diameter) to homogenize the surface, so that the CIELab parameters could be obtained: L* (brightness), a* (redness–greenness), and b* (blueness–yellowness). Also, the whiteness index (WI) of the nanoemulsion-based gels was calculated using Equation (1) [35]:

$$\mathrm{WI}(\%) = 100 - \sqrt{(100 - \mathrm{L}^{*})^{2} + \mathrm{a}^{*2} + \mathrm{b}^{*2}} \tag{1}$$

### 2.4. Rheological Properties of Nanoemulsion-Based Gels

2.4.1. Flow Properties

The flow behavior of all the NBGs was characterized using a rotational rheometer (Rheolab QC, Anton Paar, Austria) equipped with a concentric cylinder geometry (CC27, Anton Paar, Austria). The flow curves were determined by recording the shear stress values of the samples from 1 to $100\,\mathrm{s}^{-1}$ and from 100 to $1\,\mathrm{s}^{-1}$ for 120 s. The measurements were carried out at $37 \pm 1\,^{\circ}\mathrm{C}$, controlled with a Peltier system. The samples were placed in the geometry and allowed to stand for 10 min before measurement to recover their structure and reach the test temperature [36]. The experimental values of the flow curves were fitted to the Ostwald–de Waele model (Equation (2)):

$$\sigma = K\dot{\gamma}^{n} \tag{2}$$

where $\sigma$ is shear stress (Pa), $K$ is the consistency index (Pa s), $\dot{\gamma}$ is the shear rate ($\mathrm{s}^{-1}$), and $n$ is the flow index (dimensionless). Apparent viscosity at a shear rate of $10\,\mathrm{s}^{-1}$ ($\eta_{10\,s^{-1}}$) was used as a parameter for comparing the samples, since this shear rate represents the effort that is spent during the swallowing process of semisolid foods [37,38].

2.4.2. Viscoelastic Properties

Viscoelastic assays of the NBGs were carried out at $37 \pm 1\,^{\circ}\mathrm{C}$ with a controlled stress rheometer (MCR 72, Anton-Paar, Austria) equipped with a parallel-plate geometry (50 mm diameter, 2 mm gap). After loading the sample into the rheometer, it was allowed to stand for 10 min to stabilize and reach the test temperature. Small-amplitude oscillation sweeps (SAOS) were performed to analyze the viscoelastic properties of the samples. First, the linear viscoelasticity zone (LVR) was determined, where stress sweeps were conducted between 0.1 and 2000 Pa at 1 Hz. Then, a frequency sweep from 0.1 to 10 Hz was performed at 3 Pa, which was selected because it was within the LVR. Finally, the oscillatory rheological parameters (G′, G″, tan δ and complex dynamic viscosity-η*) at 1 Hz were calculated to compare the viscoelastic properties of the different NBGs. Measurements were performed in duplicate.

### 2.5. Textural Properties

Cylindrical samples of the NBGs (4 cm diameter and 2 cm height) were subjected to texture profile analysis (TPA) using a texture analysis device (Zwick/Roell Z0.5, Zwick

GmbH & CO, Ulm, Germany). For this purpose, the samples were subjected to a two-cycle compression, where in each compression cycle, the NBG sample was compressed to 20% of its original height using an aluminum cylinder probe (5 cm diameter) operated at 0.1 mm/s. After the assays, the hardness (maximum force during the first compression); cohesiveness (ratio between the areas obtained from the second and first compressions); adhesiveness (negative maximum force after the first compression); springiness (ratio between the height of maximum force during the second and first compressions); and chewiness (value obtained from hardness × cohesiveness × springiness) parameters were calculated to compare the samples.

### 2.6. Fourier Transform Infrared Spectroscopy (FT-IR)

Fresh NBGs were characterized using Fourier transform infrared spectroscopy (FT-IR) to identify possible molecular interactions between the components of the gels. The measurements were conducted using an FT-IR spectrometer with an attenuated total reflectance unit (ATR) (Diamond Two, Perkin Elmer, England). All samples were scanned in the wavenumber range of 4000–500 cm$^{-1}$ at a resolution of 1 cm$^{-1}$. Calibration was performed using a background spectrum recorded from the clean and empty cell at room temperature (25 °C). The FT-IR spectra were smoothed and baseline-corrected using the SpectrumTM 10 software.

### 2.7. Water-Holding Capacity of Nanoemulsion-Based Gels

The NBGs' water-holding capacity (*WHC*) was determined according to the methodology proposed by Liu et al. [39]. For this purpose, 20 g of each sample was placed in a 50 mL centrifuge tube and centrifuged (Universal 32R, Hettich, Tuttlingen, Germany) at 8000 rpm for 30 min at 4 ± 1 °C. Later, the water released from the sample was weighed on an analytical balance (AUX120, Shimadzu, Japan). The *WHC* of the nanoemulsion-based gels was calculated using Equation (3):

$$WHC\ (\%) = \frac{W_T - W_F}{W_T} \times 100\% \qquad (3)$$

where $W_T$ is the mass of the total water of each sample and $W_F$ is the mass of the water released after the centrifugation process.

### 2.8. Statistical Analysis

The statistical analyses were performed using two-way ANOVA (Analysis of Variance) and Tukey's test with a significance level of a = 0.05 using the XLStat© 2019 software (Addinsoft, Paris, France). The experiments were performed in duplicate, and each replica was measured at least twice (minimum of four measurements). The results were reported as the average of all the measurements and their corresponding standard deviation.

## 3. Results and Discussion

### 3.1. Visual Appearance of Nanoemulsion-Based Gels

The visual appearance of the nanoemulsion-based gels (NBGs) prepared with different types and concentrations of hydrocolloids is shown in Figure 2. All NBGs presented a homogeneous morphology, indicating that all the samples formed a stable nanoemulsion gel with no phase separation. Also, all the samples could retain their shape after gel formation (Figure 2) because of the formation of strong interactions between the different components of the NBGs.

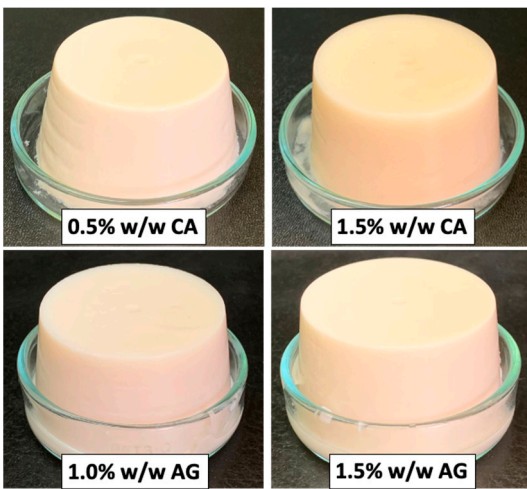

**Figure 2.** Photographs of the nanoemulsion-based gels with different types and concentrations of hydrocolloids.

### 3.2. Color Properties

Food color is traditionally represented using the CIELab color space, where L* denotes brightness, a* denotes the variation from green to red, and b* represents the variation from blue to yellow [40]. Differences between samples in the CIELab parameters were found, depending on the hydrocolloid type and concentration (Table 1). In the case of agar-based gels, no significant differences ($p > 0.05$) in the color parameters were obtained, regardless of the agar concentration. In addition, these gels showed a high whiteness index percentage (over 83%), which slightly varied with the increase in the hydrocolloid concentration (Table 1). On the contrary, the color of the κ-carrageenan-based gels differed depending on their concentration. At the lowest κ-carrageenan concentration, no differences were obtained with respect to the agar-based gels (Table 1 and Figure 2). However, after increasing the κ-carrageenan concentration from 0.5 to 1.5% *w/w*, the color of the gels changed to a more yellow color, and the whiteness index decreased significantly. These differences could be due to the intrinsic color of these hydrocolloids. According to Martín del Campo et al. [41], most types of commercial carrageenan retain a cream-yellow color after extraction and purification, with some pigments remaining in the product. For this reason, the gels with κ-carrageenan showed a more yellow color, especially at the highest concentrations.

**Table 1.** Color parameters of nanoemulsion-based gels with different hydrocolloid types and concentrations.

| Hydrocolloid | | L* | a* | b* | Whiteness Index (%) |
|---|---|---|---|---|---|
| Type | Concentration (% *w/w*) | | | | |
| Carrageenan | 0.5 | 85.2 ± 0.8 [b] | −0.52 ± 0.06 [b] | 8.19 ± 0.16 [b] | 83.05 ± 0.63 [b] |
| | 1.5 | 83.3 ± 0.3 [c] | 0.28 ± 0.05 [a] | 18.41 ± 0.45 [a] | 76.03 ± 0.10 [c] |
| Agar | 1.0 | 86.7 ± 0.2 [a] | −0.77 ± 0.04 [c] | 8.05 ± 0.16 [b] | 84.42 ± 0.24 [a] |
| | 1.5 | 86.5 ± 0.1 [ab] | −0.83 ± 0.01 [c] | 8.71 ± 0.01 [b] | 83.89 ± 0.07 [a] |

Note. Mean values with a common superscript letter do not differ significantly ($p > 0.05$, Tukey's test).

### 3.3. Rheological Characterization of Nanoemulsion-Based Gels

Rheology plays an important role in understanding structural breakdown during the oral processing of food [42], and rheological characterization could offer guidance to improve safe food consumption for people with swallowing difficulties, according to the International Dysphagia Diet Standardization Initiatives (IDDSI) [43]. For this reason, nanoemulsified gels were characterized based on their flow and viscoelastic properties to

obtain an adequate food rheology, which could be used for developing safe food products for seniors.

### 3.3.1. Flow Properties

In general, all NBGs presented a non-Newtonian and shear-thinning flow behavior (Figure 3A), showing progressive structure disruption under the application of the shear rate (decreased viscosity values with an increasing shear rate). This behavior is ideal for developing easy-to-swallow foods. since it demonstrates that a non-Newtonian behavior can render products safer to swallow than Newtonian fluids, posing a lower aspiration risk [44]. The hysteresis area between the upward and downward curves was observed in all the samples, indicating a thixotropic behavior, which was more evident at the highest hydrocolloid concentration. This phenomenon may be due to the formation of strong intermolecular forces among the hydrocolloid chains, since there are a more significant number of junction points among them, allowing for gel structure shaping. In this sense, these intermolecular interactions tend to break during the upward shear rate, and there is not enough time for a partial or total recovery of the gel structure, giving rise to the observed hysteresis [45].

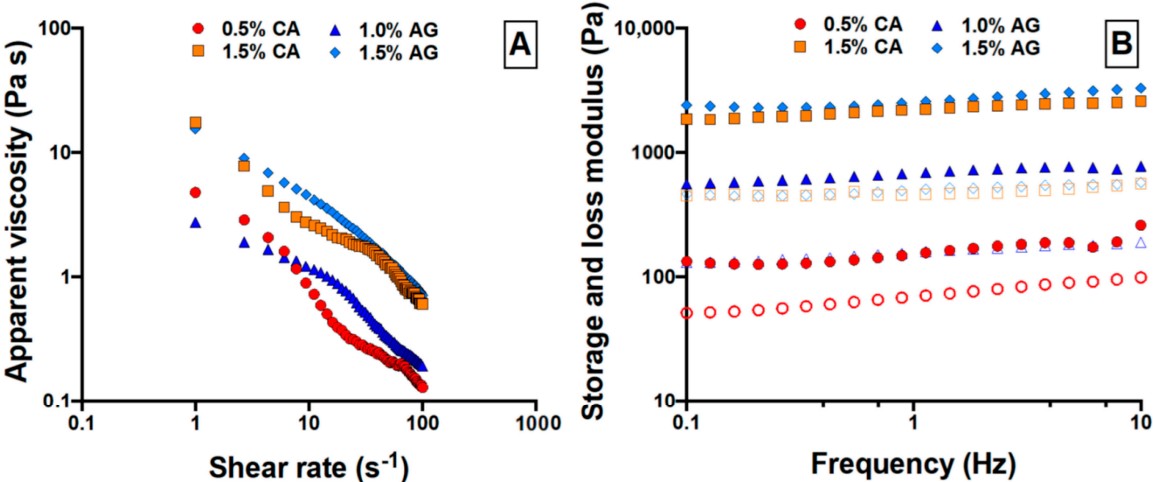

**Figure 3.** (**A**) Viscosity curves (upward and downward curves) and (**B**) mechanical spectra (G′: filled symbols, and G″: empty symbols) of the nanoemulsion-based gels with different hydrocolloid types and concentrations. CA: carrageenin, AG: agar.

On the one hand, the experimental data for the descendent flow curves of all the gels were fitted to the Ostwald–de Waele model (power law), with $R^2$ values between 0.96–0.98 and 0.81–0.92 for the agar- and κ-carrageenan-based gels, respectively. Table 2 shows the flow parameters obtained from the fitted data for all the NBGs, where significant differences (*p*-value < 0.05) were observed between samples, depending on the type and concentration of hydrocolloid. Regarding the consistency index (K), the κ-carrageenan-based gels showed the highest values, regardless of the hydrocolloid concentration. These differences could be due to the anionic sulfate content in this hydrocolloid. Carrageenan has a higher sulfate content than agar; hence, it can form firmer gels through disulfide bonding between its chains [46], which increases the gels' consistency.

**Table 2.** Rheological properties of the nanoemulsion-based gels with different hydrocolloid types and concentrations.

| Hydrocolloid | | K (Pa s) | n (-) | $\eta_{10s-1}$ (Pa s) | G′ (Pa) | G″ (Pa) | tan δ (-) | η* (Pa s) |
|---|---|---|---|---|---|---|---|---|
| Type | Concentration (% *w/w*) | | | | | | | |
| Carrageenan | 0.5 | 6.2 ± 0.2 [c] | 0.17 ± 0.01 [c] | 0.97 ± 0.02 [b] | 155 ± 9 [d] | 66 ± 5 [d] | 0.40 ± 0.05 [a] | 25 ± 1 [d] |
| | 1.5 | 16.8 ± 0.8 [a] | 0.28 ± 0.01 [b] | 3.26 ± 0.26 [a] | 2269 ± 45 [b] | 537 ± 28 [a] | 0.21 ± 0.01 [b] | 328 ± 6 [b] |
| Agar | 1.0 | 2.8 ± 0.2 [d] | 0.45 ± 0.04 [a] | 0.77 ± 0.05 [b] | 682 ± 58 [c] | 163 ± 3 [c] | 0.24 ± 0.02 [b] | 103 ± 11 [c] |
| | 1.5 | 10.0 ± 0.8 [b] | 0.43 ± 0.01 [a] | 3.02 ± 0.33 [a] | 2597 ± 38 [a] | 484 ± 24 [b] | 0.20 ± 0.01 [b] | 378 ± 11 [a] |

Note. K: consistency index, n: flow index, $\eta_{10s-1}$: apparent viscosity at a shear rate of 10 s$^{-1}$. G′: storage modulus, G″: loss modulus, tan δ: loss tangent angle, η*: complex dynamic viscosity at 1 Hz. Mean values with a common superscript letter do not differ significantly ($p > 0.05$, Tukey's test).

On the other hand, the NBGs showed flow index values < 1 (Table 2), confirming their shear-thinning flow behavior. However, significant differences ($p$-value < 0.05) in the flow index values were observed between samples, depending on the hydrocolloid type. The κ-carrageenan-based gels were more pseudoplastic than the agar-based gels (lower n values), which could be due to the structural differences between these hydrocolloids, which can be linked directly to their rheological properties [47]. In this sense, κ-carrageenan can form a helix structure, given the reduction in conformational flexibility. The latter is caused by binding between the cations of the sulfate group and anhydrous-galactosyl residue [48], where the polymeric chains can be aligned more easily with the shear rate. In turn, agar-based gels have a more complex structure due to inter- and intramolecular hydrogen bonding in the gel network, forming a tetrahedral ice-like structure [23] and decreasing their ability to align with the flow.

Finally, the apparent viscosity values, at a shear rate of 10 s$^{-1}$ ($\eta_{10s-1}$), presented differences due to the hydrocolloid concentration (Table 2). NBGs with the lower concentration (0.5% κ-carrageenan and 1.0% agar) did not show significant differences in the $\eta_{10s-1}$ values ($p$-value > 0.05). However, the $\eta_{10s-1}$ values increased ($p$-value < 0.05) with the increase in the hydrocolloid concentration, regardless of the hydrocolloid type (Table 2). This result was expected, because the concentrations of each gelling agent were selected to obtain gels with a similar apparent viscosity. In addition, according to the International Dysphagia Diet Standardization Initiative [43], all NBG textures can be classified as level 4 (as a puree), as they do not require biting, chewing, or oral preparation. This gel texture can be chewed more easily, which is a desired characteristic for the development of foods for seniors, since it makes the swallowing process more effortless and safer.

### 3.3.2. Viscoelastic Properties

Viscoelastic properties were assessed within the linear viscoelastic region (LVR) to obtain more information about the NBG structure. The mechanical spectra (viscoelastic moduli as a function of frequency) are shown in Figure 3B. The mechanical spectra were generally similar for both hydrocolloids, where the storage modulus was always higher than the loss modulus (G′ > G″) in the range of frequencies studied. This result indicates that all the samples presented a gel-like behavior due to the formation of a well-developed network in all the NBGs [49], confirming that both hydrocolloids can form an internal three-dimensional network between oil droplets [50]. Additionally, G′ and G″ were significantly affected by the hydrocolloid concentration (Figure 3B), where the NBGs with the highest hydrocolloid concentration showed higher G′ and G″ values. An increase in hydrocolloid concentration causes polymer chains to be closer, obtaining more bonds between structural elements and forming a more compact and stronger gel network [19]. However, some differences were detected between the samples. The loss modulus of the κ-carrageenan-based gels showed a slight dependency as a function of frequency (0.1–10 Hz) (Figure 3B), which suggests a weaker internal structure and, therefore, a characteristic behavior of soft gels [51]. This result suggested that the κ-carrageenan-based gels could be deformed more

easily during their consumption than the agar ones, which is a positive result, since gels with low hardness are adequate for older people [52].

Regarding the viscoelastic parameters at 1 Hz (Table 2), the agar-based NBGs were observed to show higher values (*p*-value < 0.05) for both moduli (G′ and G″) than the κ-carrageenan-based ones, indicating the formation of a more structured gel network [53]. In addition, the phase angle (tan δ, G″/G′), which provides information about the viscous modulus and the internal structure of different NBGs [54], confirmed that the κ-carrageenan-based gels corresponded to weaker-structured systems. Despite this, Suebsaen et al. [52] and Ishihara et al. [55] mentioned that tan δ values between 0.1 and 1.0 correspond to rheological criteria for the safe swallowing of foods. In this sense, all the NBGs might be suitable for safe swallowing for elderly populations (0.2 < tan δ < 0.4). On the other hand, the complex viscosity (η*) values also showed differences between hydrocolloid types (Table 2), where the agar-based gels showed the highest values compared to the κ-carrageenan-gels, confirming the results obtained for flow behavior (Table 2).

Finally, these results suggest that both hydrocolloids (κ-carrageenan and agar) can form weak nanoemulsion-filled gels, where little effort is needed to bite and chew them, easily destroying their structures during oral processing. Therefore, these gels can be used to develop food products with adequate textural properties for seniors.

### 3.4. Texture Properties

The mechanical properties of food matrices designed for people with dysphagia problems are important for a safe swallowing process, where hardness, cohesiveness, and adhesiveness textural parameters are relevant for physiological behaviors and bolus flow patterns [42].

NBG texture was evaluated through texture profile analysis (TPA), which is frequently used to mimic the chewing process by compressing the sample twice on a flat surface, simulating the first two bites during food consumption at a constant speed [56]. Figure 4 shows the TPA curves of the NBGs with 20% compression, where differences in the TPA profiles of the samples were observed according to the hydrocolloid type and concentration. The gels with κ-carrageenan required greater force for their compression than the agar gels, which also increased with a higher hydrocolloid concentration. Moreover, a low fracturability during the first compression was observed in the agar gels, since the curve showed two peaks during the first compression (Figure 4), while the κ-carrageenan gels did not show this behavior, given that only one peak was observed during the first compression (Figure 4).

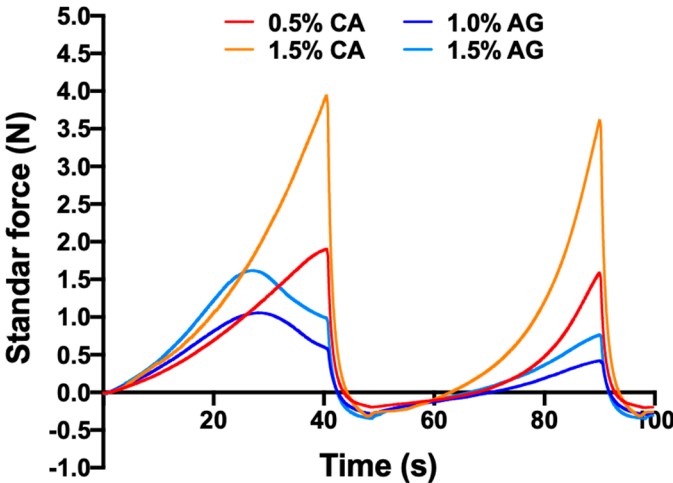

**Figure 4.** Texture profile analysis (TPA) curves of the nanoemulsion-based gels with different hydrocolloid types and concentrations. CA: carrageenin and AG: agar.

The textural parameters determined from the TPA curve were hardness, adhesiveness, cohesiveness, and springiness (Table 3), where differences between samples were observed due to the hydrocolloid type and concentration. The NBGs based on κ-carrageenan showed higher values ($p < 0.05$) of hardness than the agar-based ones (Table 3). According to Wada et al. [57], easy-to-swallow foods should have hardness values under 15,000 N/m$^2$; hence, all NBGs suit seniors with dysphagia disorders. Despite this, κ-carrageenan gels can form a more integrated gel network that increases the strength of compression because of its high molecular weight (788.7 g/mol) in comparison to agar (336.3 g/mol) [58]. Finally, this result agrees with the higher consistency index and apparent viscosity obtained for these samples (Table 2).

**Table 3.** TPA parameters of the nanoemulsion-based gels with different hydrocolloid types and concentrations.

| Hydrocolloid | | Hardness (N/m$^2$) | Adhesiveness (J/m$^2$) | Cohesiveness (-) | Springiness (-) |
|---|---|---|---|---|---|
| Type | Concentration (% *w/w*) | | | | |
| Carrageenin | 0.5 | 251.8 ± 24.5 [b] | 0.02 ± 0.002 [c] | 0.24 ± 0.02 [b] | 0.56 ± 0.01 [b] |
| | 1.5 | 438.7 ± 26.6 [a] | 0.05 ± 0.002 [b] | 0.46 ± 0.02 [a] | 0.83 ± 0.01 [a] |
| Agar | 1.0 | 152.1 ± 3.9 [b] | 0.05 ± 0.005 [b] | 0.14 ± 0.01 [c] | 0.53 ± 0.02 [b] |
| | 1.5 | 216.2 ± 16.9 [b] | 0.09 ± 0.007 [a] | 0.21 ± 0.03 [b] | 0.82 ± 0.04 [a] |

Note. Mean values with a common superscript letter do not differ significantly ($p > 0.05$, Tukey's test).

All the NBGs showed a slight adherence because the area between the two compressions was small (Figure 4). Despite this, significant differences ($p < 0.05$) between samples were observed depending on the hydrocolloid type and concentration (Table 3). In general, the agar-based gels presented higher adhesiveness than the κ-carrageenan ones. Adhesive foods are associated with an increased choking risk and require increased lingual effort to propel them into and through the pharynx [59]. Thus, low-adhesive textures could facilitate the swallowing process of older people, since they do not stick to the oral surface [60]. Also, Hadde et al. [61] mentioned that level 4 of the IDDSI for different foods with modified textures with adhesiveness values <0.5 would be safe for senior people.

Cohesiveness is a parameter related to the disintegration of gels in fragments during the chewing process, which was affected by both the hydrocolloid type and its concentration in this study. As shown in Table 3, the cohesiveness values of NBGs based on agar were lower than those of κ-carrageenan gels (Table 3), suggesting that agar gels are more brittle and can easily break during swallowing [32]. κ-Carrageenan gels, on the contrary, can maintain their gel structure during the two deformation processes. These differences could be related to the gels' skeletal structure and hardness, where the presence of disulfide bond groups in the κ-carrageenan gels made them stronger and more cohesive [62]. Despite this, all samples showed lower cohesiveness (0.14–0.46) values, consistent with parameters of the Japanese dysphagia-modified diet (values < 0.9) [63], being optimal for older people with dysphagia problems.

Finally, the springiness values were affected by the hydrocolloid concentration alone. As the hydrocolloid concentration increased, so did the gels' springiness (Figure 4 and Table 3). This behavior may be attributed to the increased number of hydrogen-bonding groups with the increasing hydrocolloid concentration, which improves the resistance of gels to deformation [64]. Springiness can reflect gel elasticity, where higher values suggest a homogeneous and well-connected gel structure [65]. In this sense, all gels presented low springiness values, especially at a lower hydrocolloid concentration, indicating that gels can be immediately deformed after the first compression into many small pieces, improving the swallowing process for seniors.

### 3.5. Fourier Transform Infrared (FTIR) Analysis

The FTIR spectra of the gels were obtained in order to understand the molecular interactions between the components of the nanoemulsion-based gels. The FTIR spectra of the canola oil, base nanoemulsion, and NBGs are presented in Figure 5. First, the canola oil spectra showed characteristics peaks (Figure 5A) that corresponded to the absorption bands of stretching ($3008$ cm$^{-1}$) and bending (rocking) ($1417$ cm$^{-1}$) vibrations of *cis* =C-H group; stretching vibrations (asymmetrical and symmetrical) of a -CH$_2$ group at $2923$ and $2853$ cm$^{-1}$, respectively; stretching vibrations of a -C=O group (ester) at $1743$ cm$^{-1}$; bending (scissor) vibrations of -CH$_2$ and CH$_3$ (-C-H) groups at $1465$ cm$^{-1}$; bending symmetrical vibrations of a -CH$_3$ group at $1377$ cm$^{-1}$; stretching and bending vibrations of -C-O and -CH$_2$- groups at $1160$ cm$^{-1}$; and bending (rocking) vibrations of a -(CH$_2$)$n$- group at $722$ cm$^{-1}$, which was also observed by Jamwal et al. [66]. In the case of nanoemulsion, the absorption peaks of canola oil were also observed in the spectrum (Figure 5A). However, a new peak at $1621$ cm$^{-1}$ was obtained, corresponding to the stretching vibrations of pea protein's amide I region (C=O and C=N), which was used as an emulsifier [67]. Also, the stretching vibrations of soy lecithin's phosphate group (PO$_2$- and *p*-O-C) were found in the absorption region between $1090$ and $850$ cm$^{-1}$ [68].

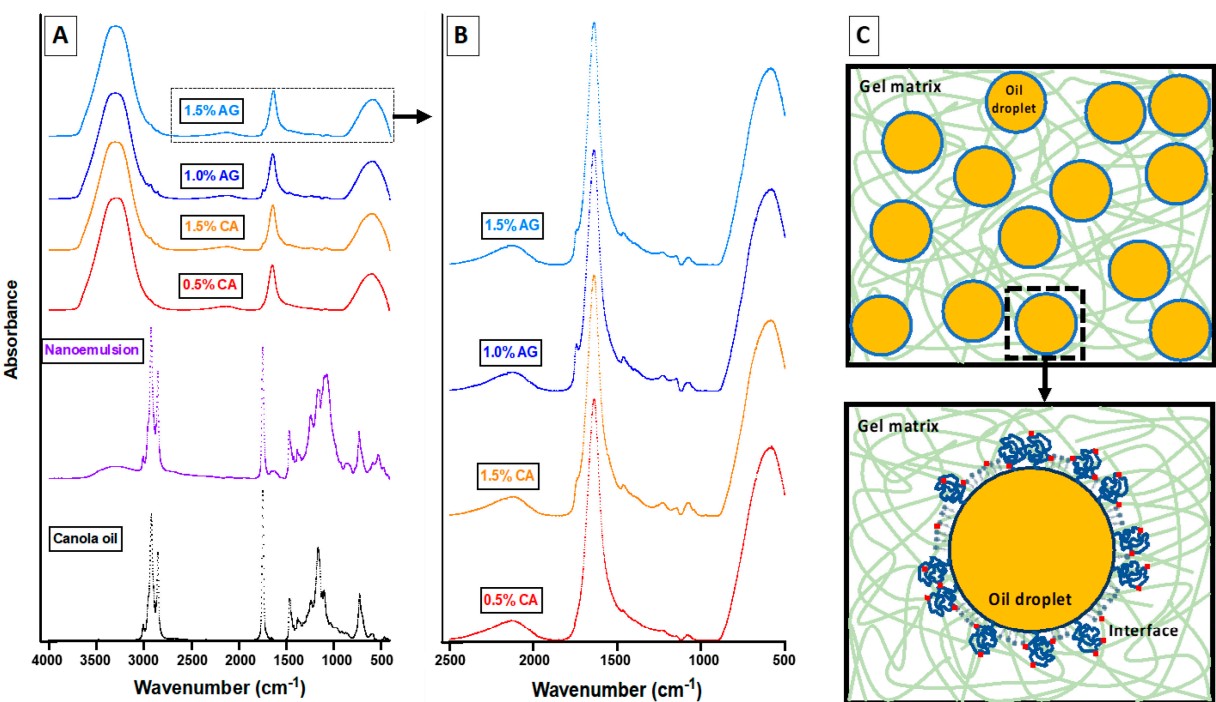

**Figure 5.** FTIR spectra from the nanoemulsion-based gels with different hydrocolloid types and concentrations (**A**,**B**) and schematic representation of nanoemulsion-based gels' structure (**C**). CA: carrageenin and AG: agar.

Different spectra were observed when all the NBGs were compared (Figure 5A), suggesting that new interactions were established between the nanoemulsion ingredients and hydrocolloids (κ-carrageenan and agar). All spectra of the NBGs revealed similar absorption peaks, independent of the hydrocolloid type and concentration. For example, the predominate absorption region appeared at around $3600$–$3200$ cm$^{-1}$ with a peak of ~$3330$ cm$^{-1}$ (Figure 5A), representing the stretching vibration of O-H groups from the water (due to the high hydration level in the sample). However, the NBGs also presented a bending vibration of O-H groups at $1638$ cm$^{-1}$ [69], indicating inter and intramolecular hydrogen bonds formed during the formation of nanoemulsion-based gels with both hydrocolloids [50]. Additionally, a new absorption region between $450$ and $650$ cm$^{-1}$ with a peak of ~$590$ cm$^{-1}$ was observed in all the samples (Figure 5A), which can be

associated with the sulfate groups of κ-carrageenan and agar [70]. Figure 5B shows a zoomed spectrum in a wavenumber range between 2500 and 500 cm$^{-1}$ for each NBG, where new absorption peaks at 2131 and 1081 cm$^{-1}$ are observed in the NBG. These peaks corresponded to the stretching vibrations of C≡C and C–O bonds, respectively, because of the molecular structure of the hydrocolloids [71]. On the other hand, the absorption peaks observed in the canola oil and nanoemulsion spectra were not distinguished in the NBG spectra, indicating the incorporation of both hydrocolloids into oil droplets through new molecular interactions. The changes in the absorption peaks of the FTIR spectra also revealed a good incorporation of both hydrocolloids into the base nanoemulsion, since non-covalent interactions such as O-H stretching were founded. Therefore, these results suggest that oil droplets present in the NBGs could act as active fillers (Figure 5C), because the interface could be connected with the gel network through emulsifiers (pea protein and soy lecithin) with non-covalent bonds [15].

### 3.6. Water-Holding Capacity

The water-holding capacity (WHC) was studied to evaluate the physical stability of the NBGs. In general, the NBGs with κ-carrageenan presented higher WHC values (75–87%) than the ones with agar (43–80%), especially at lower concentrations (Figure 6). These results can be related to NBG hardness (Table 3), since the κ-carrageenan-based gels presented the highest values, which positively affected the WHC values of these gels (Table 3). Based on this, the water-holding capacity of emulsion-based gels is related to their microstructural properties, especially the pore size and strength of the gel network [19]. Accordingly, κ-carrageenan can form a porous structure that binds water during gelation [72]. This fact means that κ-carrageenan can bind the free water molecules using hydrogen bonds due to the presence of anionic sulfate groups that improve its water-holding capacity [73]. In turn, agar-based gels are characterized by a poor WHC, since their structure is weaker and more brittle (Table 3), reducing water retention [74,75]. In addition, the concentration also has a significant effect on WHC, since the NBGs with the highest hydrocolloid concentration (1.5% *w/w*) presented a %WHC >80% (Figure 6). This behavior may be due to the gel structure becoming stronger and denser at the highest hydrocolloid concentration, retaining the water in the gel network and reducing the amount of free water after centrifugation. Also, it should be noted that the effect of concentration was more pronounced in the agar-based gels, where there was a significant increase in the %WHC (42.8% and 79.9% WHC for 1.0% and 1.5% *w/w* agar, respectively) (Figure 6). In this sense, we can hypothesize that agar-based gels might be more elastic (Figure 3 and Table 2) when their WHC is lower, because polysaccharide–polysaccharide interactions predominated in the gel network of this sample. For this reason, the polymeric chains formed by this hydrocolloid make them prone to syneresis, because they cannot hold water securely through capillary forces [19]. Instead, in the κ-carrageenan gels, the polysaccharide–water interactions could predominate due to the capability of sulfate groups to bind water molecules, as previously mentioned. These gels show lower cohesivity and a more brittle gel network than κ-carrageenan gels, promoting syneresis.

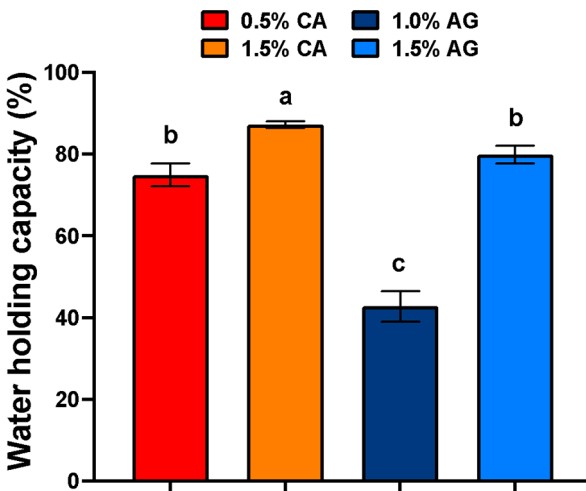

**Figure 6.** Water-holding capacity of the nanoemulsion-based gels with different hydrocolloid types and concentrations. CA: carrageenin and AG: agar. Mean values with a common letter do not differ significantly ($p > 0.05$, Tukey's test).

## 4. Conclusions

The results of this study showed that most of the physical properties of NBGs depend on the type and concentration of hydrocolloids. All NBGs presented a shear-thinning flow behavior with different levels of consistency and pseudo-plasticity. Similarly, the viscoelasticity of all the NBGs showed a predominantly elastic behavior over viscous behavior, making this effect more pronounced in the agar-based gels. Therefore, because they can be easily swallowed, it is possible to form gels with different rheological properties using agar and carrageenan as gelling agents. Regarding textural properties, the obtained NBGs were characterized by a low adhesiveness and cohesiveness, which means they could be suitable for consumption by seniors. FTIR spectra analysis revealed interactions between the components of the NBGs, suggesting that the oil droplets of nanoemulsions act as active fillers in the gel network. In addition, the water-holding capacity also depended on the type and concentration of hydrocolloid, since the NBGs with κ-carrageenan retained more water than the NBGs with agar due to the different network structure that each hydrocolloid forms. Finally, it is possible to obtain gels based on nanoemulsions with different physical properties, which could be useful for developing foods with varied textures for the senior population with swallowing problems.

**Author Contributions:** Conceptualization C.A.; methodology C.S., A.L. and N.R.; formal analysis and writing—original draft preparation N.R.; writing—review and editing C.A. and R.N.Z.; funding acquisition and project administration C.A. All authors have read and agreed to the published version of the manuscript.

**Funding:** This research was funded by Vicerrectoria de Investigación, Innovación y Creación (VRIIC, USACH, Chile) by project Dicyt-Regular No. 082171AA (Carla Arancibia) and research contract No. USA2155_Dicyt (Natalia Riquelme).

**Institutional Review Board Statement:** Not applicable.

**Informed Consent Statement:** Not applicable.

**Data Availability Statement:** Not applicable.

**Acknowledgments:** The authors thank Blumos S.A. for providing free soy lecithin and pea protein samples.

**Conflicts of Interest:** The authors declare no conflict of interest.

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
