# Peer review of "Effect of Gelling Agent Type on the Physical Properties of Nanoemulsion-Based Gels"

_colloids, doi:10.3390/colloids7030049_

Round 1

Reviewer 1 Report

My Dear,

In this submission, authors try to develop nanoemulsion based soft gels with different hydrocolloid types and concentrations of gelling agents, in order to evaluate their physical properties which could be helpful to elaborate foods with varied textures.

As a whole, the general layout visually is good to be published in the Colloids and Interfaces, but I could not find any serious and deep mechanistic discussion in the characterization sections. The context should be more detailed.

The following major revisions should be addressed before the publication of the manuscript:

1-Please highlight a statement of hypothesis on the novelty aspect and significance of this work. What is the advantages your idea in comparison with previous studies. What problems &/or gaps in scientific knowledge are attempting to be clearly addressed at the end of the Introduction. They should explain more why this topic is interesting and how this work can be useful. The potential application area of the study-i.e., the potential of nanoemulsion gels for food products also be highlighted here.

2-The mechanism of gelation and potential interdroplet interactions is not clear. The schematic figure is necessary for better understanding.

3- I could not find any deep discussions about the role of porosity on the water holding capacity of samples?  

4-They proposed some scenarios for explanation of rheological behavior of soft gel with different types and concentrations of gelling agents but they do not have any analytical and experimental evidences to confirm their finding. In my point of view. The main critical issue in this work is lack of some experimental results such as FT-IR, SEM, and porosimetry, to find that the effect of gelling agents in more details.

Author Response

  • 1-Please highlight a statement of hypothesis on the novelty aspect and significance of this work. What is the advantages your idea in comparison with previous studies. What problems &/or gaps in scientific knowledge are attempting to be clearly addressed at the end of the Introduction. They should explain more why this topic is interesting and how this work can be useful. The potential application area of the study-e., the potential of nanoemulsion gels for food products also be highlighted here.

Response: Thanks for the comment. We modified the introduction to provide our work's importance and novelty. Please see lines 30-33, 43-45, 55-65, 71-76, 90-92, 97-102, and 106-108.

  • 2-The mechanism of gelation and potential interdroplet interactions is not clear. The schematic figure is necessary for better understanding.

Response: We agree with the reviewer´s observation. We have included a new characterization based on FT-IR analysis. For this reason, FT-IR spectra of nanoemulsion-based gels, canola oil, and base nanoemulsions are in paragraph 3.5 (lines 400-444), in order to hypothesize the interactions among nanoemulsions components and hydrocolloids used as gelling agents. Besides, a schematic representation was included. Please see figure 5.

  • 3- I could not find any deep discussions about the role of porosity on the water-holding capacity of samples?  

Response: We agree with the comment, but now we don’t have the equipment to determine the porosity of samples; for this reason, we hypothesized the sample behavior according to the literature. Further, we will take it into account in future works.

4-They proposed some scenarios for explanation of rheological behavior of soft gel with different types and concentrations of gelling agents but they do not have any analytical and experimental evidences to confirm their finding. In my point of view. The main critical issue in this work is lack of some experimental results such as FT-IR, SEM, and porosimetry, to find that the effect of gelling agents in more details.

Response: Thanks for the comment. In order to explain the effect of the type and concentration of hydrocolloid in the formation of nanoemulsion-based gels, we included FT-IR measurement and their results (please see figure 5 and lines 400-444). However, as mentioned before, we don’t have the equipment to analyze other structural properties of samples.

Reviewer 2 Report

The article by Riquelme N. et al. describes the obtaining of hydrogels containing emulsified nanodroplets. The authors argue that these droplets are nano-sized and that their placement in a hydrogel matrix facilitates their use as nutrients for elders.

Above all, the English in the article is poorly understood. However, this is of no importance, since the article contains many speculations. The most important speculation is that the authors use nanoemulsion, which has never been proven. Overall, the article is trying very hard to give evidence of something that is not the case (at least in this version of the article). The article should be revised before publication.

Specific comments are as follows.

Line 24: “which could be helpful to elaborate foods with varied textures adapted for different types of population.” The authors do not make comparisons of their systems with food. The authors do not research the food preferences of different types of populations. What is the texture of food? What is the measure of texture? What is the texture of the authors' nanoemulsions? The authors should remove all meaningless speculation from the text.

Line 29: “The increase in the senior population (>60 years old) has become one of the most critical challenges for the Food Industry due to the physiological changes caused by aging (Calligaris et al., 2022).” Persons over 60 have always lived and it has not caused any "critical" problems in the food industry. This is unnecessary speculation that should be deleted. Also, the following is a description of 85+ years old rather than 60+.

Line 41: “This article until they disintegrate”. It's not clear what that means.

Line 53: “which can be greatly improved by confining oil droplets within a three-dimensional solid-like aqueous network.” Firstly, it is not clear why a nanoemulsion should be placed into a network if nanoemulsions are stable by definition. Secondly, it is not clear what an "aqueous network" is. Is it a network of water molecules? Generally speaking, emulsions consisting of large micro-sized droplets that are prone to sedimentation/creaming are required to be placed in a 3D network from a finely dispersed phase (see, e.g., 10.1021/acs.energyfuels.0c02797). This point should be addressed in the article. However, it is not clear why it is necessary to insert an emulsion of nano-droplets into the three-dimensional network, as this emulsion should be stable from sedimentation/creaming due to its small size of nanodroplets.

Line 64: “with controlled textural properties”. What are textural properties? What kind of textural properties are there? Authors should list these properties, as they operate with terms that are not widely known.

Line 76: “where agarose is responsible for gelling properties”. The authors should write about why these "gelling properties" arise.

Line 86: “easy-to-swallow soft gels”. What the yield stress and elastic modulus are for easy-to-swallow soft gels? This is very interesting to learn from the article that is being positioned that way.

Line 108: “The base nanoemulsion had a droplet size equal to 188±1 nm and a polydispersity index of 0.14, which were determined by Zetasizer (NanoS90, Malvern Instruments, UK).” The authors should provide a correlogram and particle size distribution as evidence.

Line 110: “Finally, the nanoemulsion-based soft gels (NGS) were prepared by mixing the base  nanoemulsion with hydrocolloids at two concentrations: κ-carrageenan (0.5 and 1.5% w/w) and agar (1.0 and 1.5% w/w).” The authors should provide the molecular weight of κ-carrageenan and the molecular weight of agar. In addition, it remains unclear even after the introduction: Why do we need to use polymer thickeners?

Line 113: “to obtain similar apparent viscosity”. What is that viscosity and at what rate was it measured? Why is the same viscosity needed?

Line 140: “Apparent viscosity at a shear rate of 10 s-1 was used as a parameter for compared samples”. I hope the authors know that this is a very high shear rate at which the hydrogels stop flowing and slip on the wall. The viscosity of gels should be compared at very low shear rates of about 0.001-0.01 1/s when their structure undergoes little strain.

Line 173: “The experiments were performed in duplicate, and the results were reported as the  average and their corresponding standard deviation.” The standard deviation cannot be calculated correctly with twice repeated experiments. Please remove this to avoid any embarrassment, or run a normal number of tests (more than 5).

Line 180: “All NGS presented a homogeneous morphology, indicating that all samples formed stable nanoemulsion-gel with no phase separation.” The samples are completely opaque, meaning that they have micro-sized droplets. Nanoemulsions are out of the question here. Nanoemulsions are transparent. The authors should remove speculation and the word "nanoemulsions".

Line 220, Figure 2A. This figure is uninformative. The authors should present the dependence of viscosity on shear stress instead of the dependence of stress on shear rate.

Line 269: “a gel-like behavior with an internal three-dimensional network (Domian & Szczepaniak, 2020; Espert et al., 2020)”. The references provided here contain no useful information; the authors should use additional references (e.g., 10.1016/j.carbpol.2021.118509).

Lines 312-313: “1.86Agar”. There is a typo here.

Line 314: “Cohesiveness is a parameter…, compression, … springiness, … cohesivity …”. For all these parameters, the formulas for calculations should be provided in the experimental part.

Line 380: “being suitable for consumption by the elderly”. The article should contain clear evidence that this is possible (e.g. ingestion), perhaps in the form of tables.

Author Response

Specific comments are as follows.

  • Line 24: “which could be helpful to elaborate foods with varied textures adapted for different types of population.” The authors do not make comparisons of their systems with food. The authors do not research the food preferences of different types of populations. What is the texture of food? What is the measure of texture? What is the texture of the authors' nanoemulsions? The authors should remove all meaningless speculation from the text.

Response: Thanks for the comment. We changed some phrases to avoid speculation about our results. Please see lines 23 and 487 .

  • Line 29: “The increase in the senior population (>60 years old) has become one of the most critical challenges for the Food Industry due to the physiological changes caused by aging (Calligaris et al., 2022).” Persons over 60 have always lived and it has not caused any "critical" problems in the food industry. This is unnecessary speculation that should be deleted. Also, the following is a description of 85+ years old rather than 60+.

Response: We agree with the comment. In this work, we consider that people over 60 years are part of this population group, according to the guidelines of the World Health Organization. However, we have modified the first phrase of the introduction to clarify. Please see lines 30-31.

  • Line 41: “This article until they disintegrate”. It's not clear what that means.

Response: Thanks for the observation. We eliminate the words “This article” to improve the phrase's meaning. Please see lines 43-45.

  • Line 53: “which can be greatly improved by confining oil droplets within a three-dimensional solid-like aqueous network.” Firstly, it is not clear why a nanoemulsion should be placed into a network if nanoemulsions are stable by definition. Secondly, it is not clear what an "aqueous network" is. Is it a network of water molecules? Generally speaking, emulsions consisting of large micro-sized droplets that are prone to sedimentation/creaming are required to be placed in a 3D network from a finely dispersed phase (see, e.g., 10.1021/acs.energyfuels.0c02797). This point should be addressed in the article. However, it is not clear why it is necessary to insert an emulsion of nano-droplets into the three-dimensional network, as this emulsion should be stable from sedimentation/creaming due to its small size of nanodroplets.

Response: Thanks for the comment. We understand that the paragraph is unclear, so we have modified them to clarify what nanoemulsion-based gels consist of and the study's relevance. Please see lines 60-65.

  • Line 64: “with controlled textural properties”. What are textural properties? What kind of textural properties are there? Authors should list these properties, as they operate with terms that are not widely known.

Response: Thanks for the question. We included information to clarify the textural properties meaning,  including a list of these properties. Please see lines 71-76.

  • Line 76: “where agarose is responsible for gelling properties”. The authors should write about why these "gelling properties" arise.

Response: Thanks for the marks. New information was added about the gelling properties of agarose. Please see lines 89-91.

  • Line 86: “easy-to-swallow soft gels”. What the yield stress and elastic modulus are for easy-to-swallow soft gels? This is very interesting to learn from the article that is being positioned that way.

Response: Thanks for the comment. According to the literature, the viscoelastic parameter used for criteria of easy-to-swallow soft gels corresponds to tan ? (G″/G′), where gels with tan ? values between 0.1-1.0 are suitable for safe swallowing (Ishihara et al. (2011) doi: 10.1016/j.foodhyd.2010.09.022; Suebsaen et al. (2019) doi: 10.1016/j.fbio.2019.100477). This information was added in the results section. Please see lines 335-338.

  • Line 108: “The base nanoemulsion had a droplet size equal to 188±1 nm and a polydispersity index of 0.14, which were determined by Zetasizer (NanoS90, Malvern Instruments, UK).” The authors should provide a correlogram and particle size distribution as evidence.

Response: Thanks for the comment. We included an additional figure (Figure 1) about the particle size distribution obtained for base nanoemulsion. Please see lines 129-132.

  • Line 110: “Finally, the nanoemulsion-based soft gels (NGS) were prepared by mixing the base nanoemulsion with hydrocolloids at two concentrations: κ-carrageenan (0.5 and 1.5% w/w) and agar (1.0 and 1.5% w/w).” The authors should provide the molecular weight of κ-carrageenan and the molecular weight of agar. In addition, it remains unclear even after the introduction: Why do we need to use polymer thickeners?

Response: Thanks for the comments. First, we included the molecular weight of gelling agents in the 2.1 section (materials). In addition, we improved the introduction to clarify the importance of using hydrocolloids as gelling agents in the preparation of nanoemulsion-based gels. Please see lines 61-63 and 115-116.

  • Line 113: “to obtain similar apparent viscosity”. What is that viscosity and at what rate was it measured? Why is the same viscosity needed?

Response: Thanks for the question. We included the requested information. Please see lines 135-139.

  • Line 140: “Apparent viscosity at a shear rate of 10 s-1 was used as a parameter for compared samples”. I hope the authors know that this is a very high shear rate at which the hydrogels stop flowing and slip on the wall. The viscosity of gels should be compared at very low shear rates of about 0.001-0.01 1/s when their structure undergoes little strain.

Response: Thanks for the remark. In this study, we decided to use the values of apparent viscosity at a shear rate of 10 s-1 to compare the samples since this parameter is correlated with oral shear rate during the swallowing process of semisolid foods (Chen & Stokes, 2012 doi: 10.1016/j.tifs.2011.11.006; Sharma et al., 2020 doi: 10.1016/j.foodres.2020.109275). Despite this, we will consider this comment in future works.

  • Line 173: “The experiments were performed in duplicate, and the results were reported as the average and their corresponding standard deviation.” The standard deviation cannot be calculated correctly with twice repeated experiments. Please remove this to avoid any embarrassment or run a normal number of tests (more than 5).

Response: Thanks for the observation. We clarified the number of replicates for each measurement in the Statistical analysis section. Please see lines 216-219.

  • Line 180: “All NGS presented a homogeneous morphology, indicating that all samples formed stable nanoemulsion-gel with no phase separation.” The samples are completely opaque, meaning that they have micro-sized droplets. Nanoemulsions are out of the question here. Nanoemulsions are transparent. The authors should remove speculation and the word "nanoemulsions".

Response: Thanks for the comment. According to the literature, nanoemulsions are transparent only when the droplet size is small enough compared to the wavelength of the light to avoid a strong dispersion of the light (McClements & Jafari, 2018 doi: 10.1016/B978-0-12-811838-2.00001-1), commonly under 50 nm. In our case, the base nanoemulsions presented droplet sizes near 188 nm. Therefore they showed a high opacity, affecting the appearance of gels.

  • Line 220, Figure 2A. This figure is uninformative. The authors should present the dependence of viscosity on shear stress instead of the dependence of stress on shear rate.

Response: We agree with the comment. We modified Figure 2A, including viscosity on shear rate curves.

  • Line 269: “a gel-like behavior with an internal three-dimensional network (Domian & Szczepaniak, 2020; Espert et al., 2020)”. The references provided here contain no useful information; the authors should use additional references (e.g., 10.1016/j.carbpol.2021.118509).

Response: Thanks for the comment. We modified the references used in this phrase. Also, we increased the discussion to improve the understanding of our results. Please see lines 319-326.

  • Lines 312-313: “1.86Agar”. There is a typo here.

Response: Thanks for the observation. We amended the mistake.

  • Line 314: “Cohesiveness is a parameter…, compression, … springiness, … cohesivity …”. For all these parameters, the formulas for calculations should be provided in the experimental part.

Response: Thanks for the comment. We included the calculation of each textural parameter in section 2.5. Please see lines 190-194.

  • Line 380: “being suitable for consumption by the elderly”. The article should contain clear evidence that this is possible (e.g. ingestion), perhaps in the form of tables.

Response: Thank you for the remark. As mentioned before, we included information about rheological parameters used for criteria of easy-to-swallow soft gels, which should be demonstrated that nanoemulsion-based gels are suitable for the senior population.

Reviewer 3 Report

The manuscript entitled Agar and Carrageenan as Gelling Agents for Nanoemulsion-Based Soft Gels is focused on the investigation of carrageenan and agar as gelling agents for the preparation of nanoemulsion-based gels with targeted physical properties, which could be an alternative for developing food products for the senior population.

The study is well designed, the methods are adequately described and the results are supported by high quality figures and graphs.

And that could be highly appreciated for a general physical characterisation of the obtained gel structures.

However, the authors claim in the Introduction chapter that they proposed a soft gel intended to be ingested through compression between the tongue and the hard palate until disintegration. With this regard, the authors must explain the benefit of this type of gel in comparison to common liquid or semiliquid formulations. Furthermore, the authors did not comment on the nutritional value of the gels, what kind of food is going to be delivered in this mode. Next, what is the purpose of nanoemulsification?

Somehow the intro does not correspond to the subsequent information in the article.

I have significant concerns about using the term soft gel. In pharmaceutics, soft gellatin capsules or soft gels are known. The authors must avoid using the term soft gel to prevent misunderstanding.

In summary, the authors should better motivate their research and the goal of thorough physical characterisation. Still I cannot get the point of testing viscoelastic and flow properties, as well as texture, considering the intended route of administration.

Author Response

REVIEWER 3

The manuscript entitled Agar and Carrageenan as Gelling Agents for Nanoemulsion-Based Soft Gels is focused on the investigation of carrageenan and agar as gelling agents for the preparation of nanoemulsion-based gels with targeted physical properties, which could be an alternative for developing food products for the senior population.

The study is well designed, the methods are adequately described and the results are supported by high quality figures and graphs.

And that could be highly appreciated for a general physical characterisation of the obtained gel structures.

  1. However, the authors claim in the Introduction chapter that they proposed a soft gel intended to be ingested through compression between the tongue and the hard palate until disintegration. With this regard, the authors must explain the benefit of this type of gel in comparison to common liquid or semiliquid formulations. Furthermore, the authors did not comment on the nutritional value of the gels, what kind of food is going to be delivered in this mode. Next, what is the purpose of nanoemulsification? Somehow the intro does not correspond to the subsequent information in the article.

Response: Thanks for the comment. We modified the introduction section to provide the importance of this work, including the benefits and nutritional value of using nanoemulsified gels. Also, we changed the work title to clarify the manuscript content. Please see lines 2-3, 45-46, 50-51, 56-60, 94-97, and 106.

  1. I have significant concerns about using the term soft gel. In pharmaceutics, soft gellatin capsules or soft gels are known. The authors must avoid using the term soft gel to prevent misunderstanding.

Response: We agree with the reviewer's observation. For this reason, the term "nanoemulsion-based soft gels" was modified for "nanoemulsion-based gels" or "easy-to-swallow gels" in the manuscript.

  1. In summary, the authors should better motivate their research and the goal of thorough physical characterisation. Still I cannot get the point of testing viscoelastic and flow properties, as well as texture, considering the intended route of administration.

Response: Thanks for the comment. As mentioned before, we modified the introduction section to explain the goal of the physical characterization of gels. Also, we included the relevance of testing flow, viscoelastic and textural properties of gels. Please see lines 251-257, 329-331, 345-347, 349-352, and 397-400.

Round 2

Reviewer 1 Report

Dear Editor
I think that the author tries to explain some of my questions adequately. Although they didn't consider my comment about the mechanism of gelation and potential interdroplet interactions and the role of porosity on the water-holding capacity of samples but it seems that now the modified manuscript may be accepted for publishing.

Author Response

Thanks you, no comments.

Reviewer 2 Report

The authors have made some corrections to the manuscript but have ignored critical comments. The main issue with this article is the authors' claim about nanoemulsions without convincing evidence that the droplets are undoubtedly nanosized.

So, first of all, the authors must either provide direct evidence for the nanoscale of the droplets (e.g., SEM images of the systems after freeze-drying) or remove the speculation about nanoemulsions.

Second, the authors should provide the molecular weight of κ-carrageenan and agar. 788.7 g/mol and 336.3 g/mol are not the molecular weights of polymers; these are probably the molecular weights of their monomer units, which does not provide any valuable information.

Third, the authors should provide a correlogram that was used to calculate the droplet size distribution to prove that the droplets of the base emulsion are indeed 188 nm and do not contain large drops. In addition, the authors should provide microphotographs of the base emulsion and the final systems containing different amounts of polymers. At the moment (without a correlogram and data for polymer-containing systems), the claims of 188-nm droplets and nanoemulsions are highly questionable and not acceptable for scientific publication.

Fourth, the authors compare viscosity at high shear rates from 1 to 100 1/s (Fig. 3a). These are very high shear rates at which wall slip instead of flowing is most likely to occur for gel-like emulsions. The presentation of these curves describing wall slip does not have sense, and the comparison of "viscosities" is incorrect. The viscosity of gels should be compared at low shear rates of about 0.001-0.01 1/s when their structure undergoes little strain. In addition, the authors should present these dependences in log-log coordinates, as the experimental points merge with the abscissa axis in Fig. 3a. Furthermore, the authors should additionally give the dependences of viscosity on shear stress, as these dependences allow for assessing if there is wall slip.

In addition, the authors must use MDPI guidelines for citations and references.

Author Response

Answer to reviewer comments for manuscript Colloids-2209346: “Agar and Carrageenan as Gelling Agents for Nanoemulsion-Based Soft Gels”

REVIEWER 2

The authors have made some corrections to the manuscript but have ignored critical comments. The main issue with this article is the authors' claim about nanoemulsions without convincing evidence that the droplets are undoubtedly nanosized.

So, first of all, the authors must either provide direct evidence for the nanoscale of the droplets (e.g., SEM images of the systems after freeze-drying) or remove the speculation about nanoemulsions.

Response: Thanks for the comment, but we disagree with the speculation about nanoemulsion droplet size. According to the literature, Dynamic Light Scattering (DLS) technique provides a fast and adequate evaluation of nanoemulsions' distribution and droplet size through two parameters: hydrodynamic droplet size and polydispersity index. In this way, several studies have used the DLS method to determine nanoemulsions' droplet sizes with different compositions, such as Kleinubing et al. (2022), https://doi.org/10.3390/colloids604006; Abo Enin et al. (2022) https://doi.org/10.3390/colloids6030049; Erfanifar et al. (2023), doi: 10.1016/j.foodchem.2022.134871; Gedikoglu & Erunsal (2023), doi: 10.1007/s11694-023-01855-2; Hidajat et al. (2020) https://doi.org/10.3390/colloids4010005; Guttoff et al. (2015), doi: 10.1016/j.foodchem.2014.08.087; Motta et al. (2021),doi: 10.1111/lam.13411; Qian & McClements (2011), doi: 10.1016/j.foodhyd.2010.09.017; Sun et al. (2023), doi: 10.1016/j.lwt.2023.114512; Sharma et al. (2022), doi: 10.1016/j.btre.2022.e00720; Xu et al. (2022), doi: 10.1002/jsfa.11524; Zhao et al. (2022), doi: 10.1016/j.ultsonch.2022.106195; Zhang et al. (2017), doi: 10.1016/j.lwt.2016.08.046; Zhang et al. (2022), doi: 10.1016/j.colsurfa.2022.128873; Zhou et al. (2022), doi: 10.1016/j.lwt.2021.112607. Also, other reviews about nanoemulsion characterization and application have mentioned that DLS is a suitable method for determining droplet size in these kinds of systems, for example, Silva et al. (2012), doi: 10.1007/s11947-011-0683-7; Jin et al. (2016), doi: 10.1016/B978-0-12-804306-6.00001-5; Espitia et al. (2018), doi: 10.1111/1541-4337.12405; Li et al. (2021), doi: 10.1039/D0FO02686G. Therefore, we considered that the DLS is a recognized and widely used method to determine the size of nanodroplets according to scientific evidence. In this sense, we assumed that the oil droplets of our nanoemulsions are on the nanometric scale based on DLS results, and we did not speculate about the nanoemulsion's denomination.

Second, the authors should provide the molecular weight of κ-carrageenan and agar. 788.7 g/mol and 336.3 g/mol are not the molecular weights of polymers; these are probably the molecular weights of their monomer units, which does not provide any valuable information.

Response: Thanks for the comment. Our suppliers have not provided the molecular weight of both hydrocolloids because that information is confidential or they have not it. Despite this, we can determine the molecular weight through the intrinsic viscosity, being this indirect measurement. Finally, we did not consider that this information could modify our results.

Third, the authors should provide a correlogram that was used to calculate the droplet size distribution to prove that the droplets of the base emulsion are indeed 188 nm and do not contain large drops. In addition, the authors should provide microphotographs of the base emulsion and the final systems containing different amounts of polymers. At the moment (without a correlogram and data for polymer-containing systems), the claims of 188-nm droplets and nanoemulsions are highly questionable and not acceptable for scientific publication.

Response: Thanks for the comment. As mentioned, we used the DLS method to determine the base nanoemulsion's hydrodynamic oil droplet size and polydispersity index. Figure 1 shows the droplet size distribution of the nanoemulsion base. In this case, the base nanoemulsion presented a monomodal distribution of droplet size and PdI values <0,14, indicating homogeneity of oil droplet sizes. In addition, all measurements showed good correlograms, as the report is as follows. Therefore, we considered that the results obtained represent the nano-size of our oil droplets.

Fourth, the authors compare viscosity at high shear rates from 1 to 100 1/s (Fig. 3a). These are very high shear rates at which wall slip instead of flowing is most likely to occur for gel-like emulsions. The presentation of these curves describing wall slip does not have sense, and the comparison of "viscosities" is incorrect. The viscosity of gels should be compared at low shear rates of about 0.001-0.01 1/s when their structure undergoes little strain. In addition, the authors should present these dependences in log-log coordinates, as the experimental points merge with the abscissa axis in Fig. 3a. Furthermore, the authors should additionally give the dependences of viscosity on shear stress, as these dependences allow for assessing if there is wall slip.

Response: Thanks for the comment. In this work, we used shear rates from 1 to 100 s-1 in order to simulate the shearing during oral processing, for that using an apparent viscosity at 10 s-1 to compare the samples since this shear rate represents the effort that occurs during the swallowing process of semisolid foods (Mu et al. 2022 https://doi.org10.1016/j.foodhyd.2022.107754; Sharma et al. 2020 https://doi.org/10.1016/j.foodres.2020.109275; Laguna et al. 2020 https://doi.org/10.1016/j.foodres.2020.109300; Sharma et al. 2017 https://doi.org/10.1016/j.foodhyd.2016.09.040; Ross et al. 2019 https://doi.org/10.1016/j.jfoodeng.2019.05.040). Despite this, we have modified the Fig. 3A using log-log coordinates according to reviewer suggestions.

In addition, the authors must use MDPI guidelines for citations and references.

Response: Thanks for the comment. We have modified the references according to guidelines of the Journal.

Round 3

Reviewer 2 Report

Once again, for the second time, the authors have ignored critical comments. Instead of making corrections and adding necessary information, they gave formal replies. Without correcting the previously mentioned critical points, I cannot recommend the article for publication.

Critical comments on the article are as follows.

First, the authors must either provide direct evidence for the nanoscale of the droplets (e.g., SEM images of the systems after freeze-drying) or remove the speculation about nanoemulsions.

The authors reply that they use DLS to determine the size of emulsion droplets and that this is common practice. However, that is common practice for determining droplet size in low-viscosity media, whereas the authors use polymer solutions (pea protein, lecithin), and the article does not contain information on how the authors accounted for the change in viscosity of the aqueous medium due to the addition of polymers to it. Without this information, I cannot accept the data as reliable. Moreover, the authors determine the droplet size for the base emulsion. Further, they use this emulsion to make gels and call them nanoemulsions. However, just because the emulsion had droplets of a certain size does not mean that the gels will have droplets of the same size, as it is obvious that the composition of the system has changed dramatically. This is why the authors must either provide direct evidence for the nanoscale of the droplets in gels (e.g., SEM images of the systems after freeze-drying) or remove the speculation about nanoemulsions.

Second, the authors should provide the molecular weight of κ-carrageenan and agar.

The authors answer that they do not know this information and that it cannot change their results in any way. This statement is fundamentally wrong. The molecular weight of polymers is critical for the properties of their solutions and gels. Furthermore, any scientific article must contain exhaustive information to allow anyone to reproduce it precisely. Without knowing the molecular weight of polymers, it is impossible to reproduce this work.

Third, the authors should provide a correlogram that was used to calculate the droplet size distribution to prove that the droplets of the base emulsion are indeed 188 nm and do not contain large drops. In addition, the authors should provide microphotographs of the base emulsion and the final systems containing different amounts of polymers. At the moment (without a correlogram and data for polymer-containing systems), the claims of 188-nm droplets and nanoemulsions are highly questionable and not acceptable for scientific publication.

Instead of providing experimental correlograms, the authors respond "All measurements showed good correlograms." Well, then show it, at least in the supplementary materials. Without providing correlograms, I do not believe that “the base emulsion is indeed 188 nm and does not contain large drops”.

Fourth, the authors compare viscosity at high shear rates from 1 to 100 1/s (Fig. 3a). These are very high shear rates at which wall slip instead of flowing is most likely to occur for gel-like emulsions. The presentation of these curves describing wall slip does not have sense, and the comparison of "viscosities" is incorrect. The viscosity of gels should be compared at low shear rates of about 0.001-0.01 1/s when their structure undergoes little strain.

The authors reply that they used high rates to simulate "oral processing." Okay, let it be so, but the authors should show the data at low shear rates. In addition, the authors should then show the viscosity versus shear stress to be sure there is no wall slip in the measurement. Without these tests, I do not believe that the authors measured the viscosity of the gels, whereas it is more likely that the gels were slipping on the wall of the rheometer measuring unit.

Since the authors have demonstrated that they are not going to correct their article, I cannot recommend even its revision.

Author Response

Dear reviewer, we have not ignored your comments. We have tried to respond to every comment according to our possibilities.

We have provided evidence that the DLS technique is widely used for droplet size characterization (this point was clarified in the last revision). Although we can understand your comment about using TEM or SEM to determine droplet size, we considered the other techniques useful for droplet characterization. Unfortunately, we do not have the SEM or TEM equipment to analyze the particle size of the gels. Therefore, we cannot achieve your requirements

Despite this, we considered that the relevance of this work is the physical characterization of gels (optical, flow, viscoelastic, and textural properties, molecular interaction by FT-IR, and physical stability by water-holding capacity), which each one has been well developed and discussed. For this reason, we modified the title to clarify the relevance of the work.
